# “It Is Definitely a Good Program for Everyone from Every Community”: A Qualitative Study of Community Partner Perspectives on the Culturally and Linguistically Diverse (CALD) Mindfulness Program

**DOI:** 10.3390/ijerph20166608

**Published:** 2023-08-18

**Authors:** Ilse Blignault, Hend Saab, Hanan Youssef, Heba Baddah, Klara Giourgas, Lisa Woodland

**Affiliations:** 1Translational Health Research Institute, Western Sydney University, Penrith, NSW 2751, Australia; 2Multicultural Health Team, South Eastern Sydney Local Health District, Darlinghurst, NSW 2010, Australia; hend.saab@health.nsw.gov.au (H.S.); heba.baddah@health.nsw.gov.au (H.B.); klara.giourgas@health.nsw.gov.au (K.G.); 3Health Equity, Promotion and Prevention Service, South Eastern Sydney Local Health District, Darlinghurst, NSW 2010, Australia; lisa.woodland@health.nsw.gov.au; 4Centre for Primary Health Care and Equity, UNSW Sydney, Sydney, NSW 2052, Australia

**Keywords:** primary health care, equity and access to health services, health promotion, mindfulness-based intervention, community engagement, evaluation, migrant, refugee, asylum seeker, mental health

## Abstract

Meeting the health needs of migrant and refugee communities is crucial to successful settlement and integration. These communities are often under-served by mental health services. Previous research has demonstrated the effectiveness of a group mindfulness-based intervention tailored for Arabic and Bangla speakers living in Sydney, Australia. This study aimed to explore community partner perspectives on the program’s impact, contributing factors and sustainability, and to elicit suggestions for future development. Data were collected via semi-structured telephone interviews with a purposively selected sample of 16 informants. Thematic analysis was conducted using the Rigorous and Accelerated Data Reduction (RADaR) technique. Community partners welcomed the emphasis on promoting wellbeing and reported that the community-based in-language intervention, in both face-to-face and online formats, overcame many of the barriers to timely mental health care for culturally and linguistically diverse (CALD) communities, with a beneficial impact on group participants, program providers, partner organisations and the broader community. Positive outcomes led to stronger community engagement and demand for more programs. For group mental health programs, both trust and safety are necessary. Relationships must be nurtured, diversity within CALD communities recognised, and projects adequately resourced to ensure partner organisations are not overburdened.

## 1. Introduction

Modern Australia is a multicultural nation [1]; since WWII it has attracted migrants and refugees from all over the world [2]. Mental health is a national priority [3]. Despite evidence pointing to relatively high rates of mental ill health among some culturally and linguistically diverse (CALD) communities compared to the general population, they are less likely to seek out and engage with available mental health services [4]. Commonly reported mental disorders include depression, anxiety and post-traumatic stress disorder (PTSD), with rates of PTSD being particularly high among refugees and asylum seekers [4]. Commonly reported barriers to service access include language (low English proficiency), cultural and religious beliefs, stigma surrounding mental health and limited understanding of the Australian health system [5,6,7]. The evidence-based intervention discussed here, known as the Culturally and Linguistically Diverse (CALD) Mindfulness Program, was designed to improve equity and access to primary mental health care through community engagement, specifically working with community partners to deliver culturally and linguistically tailored group mindfulness-based interventions (MBIs) [8,9].

Community engagement has been variously defined, e.g., [10,11]. According to the World Health Organization (WHO), it is “a process of developing relationships that enable stakeholders to work together to address health-related issues and promote well-being to achieve positive health impact and outcomes” [11] (p. 12). Successful relationships are characterised by mutual respect, trust and a shared sense of purpose [12]. Enabling factors for effective community engagement include governance, leadership, decision-making, communication, collaboration and partnership and resources [12]. The benefits of intersectoral collaboration and partnership between government services, non-government organisations and the local community have been well documented in health promotion [13,14,15]. Ongoing partnerships ensure that community priorities and values continue to shape services and systems [10].

The WHO has identified four general approaches to community engagement in the context of health and wellbeing, based on increasing levels of community involvement: (1) community-oriented, (2) community-based, (3) community-managed and (4) community-owned [12]. The CALD Mindfulness Program (the Program) is an example of a community-based approach in which the community is consulted and involved to improve access to health services and programs by locating interventions inside the community with some external support. The Program, which is delivered at community venues, considers four socio-ecological levels of influence described in the *Ottawa Charter for Health Promotion*: developing personal skills, strengthening community action, creating supportive environments and reorienting health systems [16].

Australian population health data indicate that migrants born in Southern and South-East Asia and the Middle East and Africa are particularly likely to experience high levels of disability due to psychological distress [17]. Migrants born in Lebanon reported rates of psychological distress double that of the Australian-born population [18]. Of the 58% of Arab Australians (first and second generation) who expressed concerns about their mental health in an online survey, only 18% saw a mental health professional [19]. Among a volunteer sample of Iraqi refugees attending English language classes, 40% reported severe psychological distress and 31% reported clinically significant PTSD symptomatology, but only 33% of the latter group had ever sought mental health care [20]. Such high levels of distress, combined with low health service utilisation, highlight the need for interventions targeting these communities.

Designed as a low-intensity mental health intervention, the Program aims to reduce psychological distress and enhance community wellbeing. The Program has three components, including in-language resource development and training in MBIs for bilingual mental health clinicians and community workers. However, the foundation is a 5-week face-to-face mindfulness skills development program delivered in-language, initially to Arabic- and Bangla-speaking groups [9]. During the COVID-19 pandemic, when face-to-face groups could not be safely held, a 4-week online stress management program and a one-off online session on stress management were introduced. Today, bilingual facilitators and community workers conduct groups in a variety of languages (Arabic, Bangla, Cantonese, Greek, Mandarin, Nepali, Russian, Spanish and English for mixed groups) and various formats to cater to community needs and preferences.

Previous articles have reported on the cultural acceptability, clinical utility and effectiveness of the Program and its associated audio-visual resources [8,9,21], including adjustments made in response to the COVID-19 pandemic [22]. This innovative community-based program, with its emphasis on promotion of mental health and wellbeing, sits within Australia’s stepped care model for primary mental health care [22]. This article reports on another aspect of the ongoing program evaluation. The current study used qualitative methods and semi-structured interviews to explore community partner perspectives on the Program’s impact, contributing factors and sustainability, and any population groups that missed out, and to elicit suggestions for future development.

## 2. Materials and Methods 

### 2.1. Context

Australia’s health system is underpinned by Medicare—a universal health insurance scheme that has been in place since 1984. Responsibility for funding, operating, managing and regulating the health system is broadly shared by the Australian, State and Territory governments. Health services are delivered by a range of health professionals working in a variety of settings across the public and private health sectors [23]. Primary Health Networks are coordinating bodies that work directly with general practitioners, other primary health care providers, hospitals and the broader community to increase the efficiency and effectiveness of health services and improve the coordination of care for patients moving between different services or providers, particularly those at risk of poor health outcomes [23]. There are 31 Primary Health Networks nationally, funded by the Australian Government [23].

Central and Eastern Sydney Primary Health Network (CESPHN) provides services for 1.5 million individuals living in Australia’s largest city. CESPHN commissions services to meet population health needs and reduce barriers to access for communities with the highest needs. The region is culturally and linguistically diverse: 41% of residents are overseas born, 37% speak a language other than English at home, and 6% do not speak English well or at all [24].

CESPHN’s boundaries align with those of South Eastern Sydney Local Health District (SESLHD) and Sydney Local Health District (SLHD), which are part of NSW Health (state government). Following two positive evaluations with Arabic speakers [8,9], CESPHN commissioned the SESLHD Multicultural Health Service to deliver the group mindfulness program to CALD communities in their region. The Multicultural Health Team, which falls under the SESLHD Health Equity, Promotion and Prevention Service, works towards equitable health outcomes for people from CALD backgrounds through community projects and grants, support for health professionals and research. 

### 2.2. Program Partners

The Program was led by the SESLHD Multicultural Health Team in partnership with SESLHD and SLHD Mental Health Services, CESPHN, Western Sydney University and community partners. The latter included four settlement services, two women’s organisations, two multicultural health services, NSW Refugee Health Service and four individual partners recruited, trained and contracted to support the delivery of the Program in their community. Direction and oversight were provided by a steering committee that meets quarterly and is chaired by the Director of the SESLHD Health Equity, Promotion and Prevention Service (LW), with the Program’s clinical lead (HS) responsible for clinical governance.

### 2.3. Target Communities

The Program targeted Arabic and Bangla speakers in the first instance. Arabic speakers have a significant and long-standing presence in the region, and a well-established social infrastructure, whereas Bangla speakers constitute one of the new and emerging communities; both are considered as vulnerable and priority populations [25]. The Australian Arabic-speaking population is comprised of numerous cultural and ethnic communities, and both Muslims and Christians [2]. The Bangla population is more homogeneous (predominantly from Bangladesh and Muslim) and younger, with many growing families lacking extended family and community support. While most Bangla speakers enter Australia under the Skilled Migration Program, since 1970, successive waves of Arabic speakers have arrived under the Refugee and Humanitarian Program [2]. 

### 2.4. Program Delivery

The Program commenced in March 2017; the content of the multi-session groups and cultural adaptation of the mindfulness resources is detailed elsewhere [8,21,22]. Sessions were co-facilitated by bilingual psychologists and trained community workers who emphasised the cultural and spiritual relevance of mindfulness practice in their explanations and examples, tailoring the content according to the needs of each group (e.g., young mothers, older women or men). Previous studies demonstrated compatibility with participants’ cultural and religious values and practices and way of life [8,9,21].

By late November 2021, when interviews were conducted with community partners, 38 face-to-face mindfulness skills programs (24 Arabic and 14 Bangla) had been delivered, with a total of 459 participants. During Sydney’s first COVID-19 lockdown in 2020, program content was modified for online delivery. The resulting 4-week stress management programs (4 Arabic, 3 Bangla and 1 presented in English to a multicultural group) attracted a total of 78 participants. The second lockdown in 2021 prompted a series of one-off stress management sessions to provide mental health support to the wider CALD community. Nineteen sessions were delivered in a variety of languages (Arabic, Bangla, English, Cantonese, Greek, Russian and Spanish) to 166 participants. 

The clinical lead (an English/Arabic-speaking psychologist) was facilitator for most of the multi-session groups. An English/Bangla-speaking psychologist facilitated the remaining multi-session groups. The co-facilitators comprised English/Arabic-speaking psychology interns (including HY and HB) and bilingual community workers (including community organisation staff and individuals). Bilingual community workers who did not co-facilitate also attended the group sessions, providing language and practical support when necessary. The community partners were responsible for promoting the program and recruiting group participants. For face-to-face groups, partner organisations booked the venue and arranged refreshments and childcare. For online groups, the bilingual community workers provided technical support. All groups were free of charge. Referrals were managed by the group facilitators [9].

### 2.5. Participant Outcomes

Our first evaluation was a mixed-methods effectiveness study centred on participant mental health outcomes and experiences [9]. Based on the first 23 of the 5-week face-to-face mindfulness programs (15 for Arabic Speakers and 8 for Bangla speakers, a total of 271 participants), it demonstrated clinically and statistically significant improvements in mental health assessed on the Depression, Anxiety and Stress Scale (DASS-21) and Kessler Psychological Distress Scale (K10). Thirty new referrals were made for specialist mental health care. Participant-reported benefits included skills development and personal growth, increased ability to cope with ongoing stressors and improved wellbeing and relationships [9]. The 4-week online stress management programs were also successful, with a significant reduction in psychological distress in both language groups [22].

### 2.6. Evaluation 

#### 2.6.1. Design

In this qualitative study, we sought to understand the Program’s impact and implementation from the perspective of the community partners. Our interpretive framework was based on pragmatism, which has a focus on the research outcomes rather than the methods [26], making it especially useful for evaluation and applied research concerned to find solutions to real world-problems [27]. We adopted an appreciative inquiry approach [28] and a socio-ecological lens [29] to explore both outcomes and process. Appreciative inquiry is a positive, strengths-based approach to evaluation [28]. A socio-ecological lens considers the individual and their environment (e.g., family, community and society) [29].

#### 2.6.2. Researcher Backgrounds and Roles

Research team members have clinical and research experience in the fields of mental health and public health. HS, HB and HY (psychologists) were all directly involved in program delivery at some point. KG (health program manager) provided research administration support. IB (psychologist and lead researcher) and LW (health service director) both have broad experience in multicultural health service and policy development.

#### 2.6.3. Sample

The purposively-selected sample of community partners comprised bilingual mental health clinicians and community workers who had been involved in one or more of the face-to-face or online group programs for the Arabic or Bangla-speaking communities in the previous two years (i.e., 2020–2021). Potential informants were invited by an email from the clinical lead to take part in a 30-min telephone interview at their convenience; it was emphasised that participation was voluntary.

#### 2.6.4. Data Collection

Interviews were conducted by a project officer recruited for this purpose (HY, who was involved in development of the English language resources and co-facilitated five of the early Arabic face-to-face groups). The semi-structured interview guide, which was piloted with one of the bilingual community workers, contained 12 open-ended questions (e.g., “Can you comment on the CALD Mindfulness Program as a community wellbeing initiative?” and “What do you think is required to sustain such a program in the community?”). The full list of questions can be found in Appendix A. Questions were emailed to participants in advance and verbal consent obtained at the start of the interview. The interviews were not recorded; however, the project officer took extensive notes that informants were given the opportunity to review. 

#### 2.6.5. Analysis 

Responses to questions which asked about their contribution to the conduct of the mindfulness groups and the challenges they encountered were analysed and reported as part of the earlier evaluation [9]. Subsequently, responses to the remaining questions were examined according to the Rigorous and Accelerated Data Reduction (RADaR) technique, which uses a team-based approach and general-purpose software to code and analyse qualitative data [30]. Steps 1 and 2, ensuring that all interview notes are formatted similarly and transferring the data into one large data table, were completed by HS and checked by HY. Steps 3 and 4, successively reducing the data to produce more data tables with a focus on identifying themes (summaries) in relation to each interview question, were carried out by HS and HY under the supervision of IB. Step 5, drafting the manuscript based on the final data tables, was led by IB. Between steps, the analysis was discussed at research team meetings. Tables were created in Microsoft Word. Our orientation to the thematic analysis was primarily semantic in that we relied on the explicit content of the data to develop codes and themes [31]. We combined inductive and deductive approaches, guided by the data and bringing a socio-ecological lens [31]. The responses of Arabic and Bangla informants were compared, and references to the clinical lead and trust/safety were coded across the entire data set. Care was taken to ensure that the analysis accurately represented the views of the informants.

#### 2.6.6. Rigor

Credibility was supported through purposive sampling to ensure a cross-section of community partners with recent program experience, and the inclusion of all informant voices in the write up. HS and HY maintained an audit trail documenting partner selection, data collection and analysis steps (dependability). The whole team met regularly to reflect on each aspect of the research including the data analysis. Throughout the research process, we considered the impact of existing relationships between the research team members and community partners, including between the clinical lead and project officer and informants, and our own ideas and practices. Details of the study context, setting and informant characteristics are provided so that others can judge transferability of the findings [32]. We used the consolidated criteria for reporting qualitative research (COREQ) checklist as a guideline for reporting [33]—See Appendix A.

## 3. Results

### 3.1. Informant Characteristics

We identified 17 people who met the eligibility criteria from the 13 community partners (organisations and individuals) and were able to interview 16 of them: two clinician facilitators and 14 community workers, 9 of whom acted as co-facilitator. Ten informants (one individual partner and nine community workers) spoke Arabic, four (three individual partners and one community worker) spoke Bangla, one spoke Nepali and one spoke English only (see Table 1). Twelve informants had attended training on MBIs for CALD communities. 

### 3.2. Perceived Impact

#### 3.2.1. Impact on Group Participants

The impact of the Program was seen by community partner informants as positive and lasting, even *“life changing”* (Informant 1, I.1). In addition to improved mental health, participants were empowered through the acquisition of knowledge and skills that were relevant to their everyday needs and congruent with their cultural and religious beliefs. The expertly facilitated groups provided a safe space which enabled them to connect with others and share and discuss a highly stigmatised topic. Informants reported *“lots of positive feedback”* (I.9) and that *“the program increased the resilience of the participants*” (I.2). One informant reported that *“one of the participants said she had been considering self-harm and this completely changed her attitude about her wellbeing and her life”* (I.8). Another informant described the program as *“like a candle in the dark”* (I.15).

#### 3.2.2. Impact on Community Partner Providers 

For informants themselves, the experience was rewarding both professionally and personally, as well as *“enlightening and inspiring”* (I.12). Community partner providers reported becoming more mindful, focused and able to better manage their stress levels to prevent burnout. Self-care skills, self-reflection and awareness of their thinking style enabled them to better cope with personal grief and loss during the COVID lockdowns. One informant said, *“I had been seeing therapists before … but nothing they said really resonated with me. I think it had to do with my religious and cultural background.”* (I.5). Another explained that being able to incorporate Islamic principles and practices *“really helped me”* (I.10). Informants shared and practiced mindfulness with their families, as well as applying the mindfulness tools and other new skills (e.g., online program delivery) in other programs. Community workers developed a better understanding of how to navigate the mental health system and follow up on referrals. The two clinicians learned new mindfulness techniques and extended their community and professional networks; one remarked *“for me it was a new experience running groups using Zoom”* (I.7). All informants expressed an interest in continuing involvement with the program.

Participant recruitment was more challenging for the Bangla informants, three of whom were individual partners, as they lacked the community connections and trust of partner organisation staff. Additionally, they lacked an existing member/client database and connections with other services, and ready access to resources such as a work phone, car and venue. Thus, they had to rely, especially in the early stages, on using social media and attending local schools and community events to promote the program and recruit participants. One co-facilitator involved in several of the one-off sessions commented on the large amount of time involved, saying *“I was feeling tired every week, recruiting new people and following up, making time sheets and attendance sheets. I also had to send them text messages. It takes longer than 2 h. Every week, from my point of view, I found it challenging”* (I.6). One of the Arabic informants stated that *“it was time consuming, but we were still happy to do it”*, noting *“a lot of time is spent completing forms”* (I.3).

#### 3.2.3. Impact on Community Partner Organisations

The Program’s impact on community partner organisations and their staff was seen as *“very positive because it demonstrated that we actually cater for a person’s holistic needs … also addressing their wellbeing”* (I.1). One informant explained that it showed the organisation *“as being supportive and understanding of deeper community needs such as mental health, self-care … something we haven’t touched on before”* (I.8). Another said that it provided the organisation with an effective *“model”* to promote mental health in the CALD community and advocate for access to relevant services (I.12). The capacity building component of the program trained and upskilled staff and gave them *“powerful tools”* for addressing common mental health issues (I.14). This encouraged self-care practices and created a supportive work environment which benefited the whole organisation: *“We are human too and have to take the time to care for ourselves so that we can properly serve those in need”* (I.10). Mindfulness concepts and skills have been introduced into other organisation programs such as pain management.

#### 3.2.4. Impact on Community

The Program’s impact on the community was also seen as positive: *“It is definitely a good program for everyone from every community”* (I.6). In addition to promoting mental health awareness and literacy and facilitating access to specialist mental health services (particularly psychologists), the program introduced new vocabulary and encouraged community dialogue, helping to *“change the narrative around mental health”* (I.1), and acted as a *“soft entry point”* for participants to access other support including financial, legal or housing (I.10). In the groups, people *“become friends and make a network of support for themselves”* (I.13). Once established, the program created its own demand. Community organisations found their connections with the community strengthened through the program; however, it was observed that community members who lived outside the catchment area felt *“abandonment”* (I.7) and *“alienated”* (I.5).

### 3.3. Contributing Factors

According to informants, the main factors that contributed to the positive impact, beyond the Program itself, were related to the program providers and the community organisations involved. Trust and safety were frequently mentioned. Having a well-designed in-language program with written, audio and video resources supported participants’ mindfulness practice outside the sessions and reinforced their learning; being delivered in community language was *“very powerful”* (I.14). Group homogeneity (recruiting participants of similar age and background) supported sharing of personal experiences and their normalisation. Positive outcomes and word-of-mouth communication created demand and facilitated recruitment: *“It has had a good impact on our clients and they are enthusiastic to join”* (I.2). Adapting the program to an online format with a focus on stress management in response to the pandemic allowed people to participate *“from their home rather than having to travel”* (I.7).

All informants acknowledged the clinical lead’s knowledge, experience, and clinical and cultural competency as a significant contributory factor to the program’s success. As the main facilitator (leading multi-session groups in Arabic, Bangla and English), her ability to engage participants and to integrate mindfulness concepts with their cultural and religious beliefs and relate them to their everyday practices *“resonates really well with the group”* (I.5). One informant recounted an example of how *“someone who was sceptical initially was able think beyond the religious origins and to focus on his greater health”* (I.2). Another informant highlighted her *“authenticity and sincerity”,* saying *“we have worked with many mental health providers in the past and we can see the difference in the way [she] delivers this program and how she empowers people to be leaders”* (I.12). 

The facilitators, all experienced clinicians, created a nonjudgmental and inclusive space in which participants *“felt comfortable to share their personal issues and problems in front of everyone”* (I.4). The presence of a familiar person as co-facilitator assisted in creating a supportive environment. Between the weekly sessions, a phone call from the co-facilitator helped to maintain motivation and encouraged mindfulness practice. When paired with the non-Bangla clinical lead, the Bangla co-facilitators provided both language and cultural support. Informants who were also case workers supported the referrals to mental health and other services.

Alignment between the community partner organisation’s vision for the wellbeing of the CALD communities and the program goals strengthened intersectoral collaboration and helped facilitate the program’s favourable outcomes. Organisations provided local knowledge and practical support and promoted the program to their own clients and the broader community. One informant explained, *“We had a very good relationship and good communication with the organisers of the program, and we were able to get the program to suit the local community here”* (I.11). Another stated, *“People trusted that [our] organisation was not going to run any program that would not benefit them. That trust is what made it very successful”* (I.16). 

### 3.4. Community Wellbeing

As a community wellbeing initiative, the Program was regarded as very effective: a free, locally available, in-language program catering for a range of ages, cultural and religious backgrounds and migration experiences, presented by a trusted team in a safe environment. Informants described it as *“really useful and very valuable … really needed”* (I.3) and *“very culturally competent and culturally inclusive”* (I.12), commenting that it was *“received very well”* (I.2) by the initial target communities (Arabic and Bangla speakers) as it filled a gap in services, especially during the lockdowns. Nevertheless, *“there are so many people looking for additional support”* (I.5) and *“there are a lot more core community needs that need to be addressed”* (I.1). One informant observed, *“For us to create a healthy community it is crucial that we increase the number of healthy minds… Both men and women benefitted themselves, both mentally and physically, and this has improved overall wellbeing of the community”* (I.13). Another considered it an *“eye opener”* in highlighting the importance of wellbeing and health for the community as well as individually (I.11).

In raising mental health awareness, reducing stigma and facilitating access to relevant health and community services, the Program was considered especially beneficial for new and emerging communities and refugees with trauma backgrounds: *“Refugees need these sessions when they are newly arrived and they can know where to go and who can help them, and address mental health issues which they face, from the very beginning”* (I.4). Sections of the target communities that were identified as missing out included men (most groups were for women), adolescents, and older people and others with mobility issues or lacking internet skills. Competing work, study or family commitments were an obstacle to participation. 

### 3.5. Suggestions for Improvement

Informants suggested connecting with influential community and religious leaders; partnering with other health, disability and youth services, and settlement services; and offering the program on different days and times (including weekends and evenings) and in different formats (face-to face, online and hybrid). Some mentioned promotion, e.g., in-language advertising fliers and social media, *“even through WhatsApp”* (I.16). Others noted the need for additional groups for those who missed out and for refugees, with ongoing refinement informed by participants’ feedback and suggestions. Five weeks was considered a suitable length for the face-to-face program, being *“long enough to ensure that people remain interested and engaged”* and allow *“people to reflect”* (I.12). A day retreat was suggested for those who cannot commit to five weeks. Other suggestions for improvement included increasing session time to allow for more discussion, providing a summary at the end of each session, and offering booster sessions to reinforce skills learnt and encourage practice. In terms of content, one informant mentioned self-care, self-compassion and kindness, saying that *“people really related to this information and would like to hear more about it”* (I.6). Another proposed more psychoeducation, e.g., *“a very brief slide on anxiety and clinical depression”* (I.9). There was strong support for extending the program to neighbouring regions, even making it state-wide, and for developing similar in-language programs for other CALD communities.

### 3.6. Sustainability

All informants wanted to see the Program continued, with one asserting *“It is very important to sustain the program because of the community need”* (I.14). To build on achievements to date, they suggested booster or follow up sessions (as was done during the pandemic) and conducting regular programs in the community to keep it in the forefront of the community’s mind, *“at least twice a year”* (I.2). In addition to continued collaboration and partnership with community leaders and organisations that work with CALD communities, many talked about further workforce capacity building and delivery in more languages: *“having more people to facilitate … this should be run more regularly than just once a year to meet the community wellbeing needs”* (I.1). Informants also brought up financial support for community partner organisations to cover staff time and program expenses such as venue hire, childcare activity materials and *“community specific food for morning tea”* (I.2). One stated, *“I think the government should look at this program as a priority, especially when we have so much interest”* (I.15).

## 4. Discussion

In multicultural nations such as Australia, meeting the physical and mental health needs of people with migrant and refugee backgrounds is critical to successful settlement and integration [34]. The *NSW Plan for Healthy Culturally and Linguistically Diverse Communities 2019–2023* calls for an equitable health system where cultural and linguistic needs are recognised in policy development, service planning and delivery [35]. Across the world, primary health care (the first contact an individual with a health problem has with the health system) is regarded as “the most inclusive, equitable, cost-effective and efficient approach to enhance people’s physical and mental health, as well as social well-being” [36]. Interventions based in hospitals and health services are less likely to reduce health inequity and can further marginalise vulnerable groups [12]. Stigmatised issues such as mental health require special consideration [37,38], particularly when working with CALD communities [7,39]. Since 2017, the CALD Mindfulness Program has provided hundreds of people from Sydney’s Arabic and Bangla-speaking communities with access to culturally tailored high-quality mental health care and information, simultaneously building the capacity of program providers and partner organisations and reducing mental-health-related stigma [9]. This qualitative study, conducted when the Program had been operational for four-and-a-half years, sought to understand how it was perceived by the community partners. 

SESLHD Multicultural Health Team has a long history of CALD community engagement, supported by a multicultural grants program that ran from 2000 to 2020. The “Healthy Communities” grants category was open to community organisations wanting to work in partnership with SESLHD to deliver programs and services to enhance the health of CALD communities. For each mindfulness group, activities and processes were informed by in-depth understanding of the local community, their priorities, and their cultural and religious values. In addition to language, recruitment took into account social circumstances and acculturation. Most groups were women-only, given gender sensitivities. Among the Arabic-speaking participants, psychological distress was often related to refugee experience (past or recent), war trauma, domestic violence and relationships issues, parenting teenagers or caregiver burden. For the Bangla speakers, psychosocial triggers were often associated with settlement, e.g., family separation, social isolation, adjustment issues, learning English and employment. Despite these differences, community partner informants perceived the impact on group participants and the broader community as similar. For the Multicultural Health Team, the lack of established Bangla community infrastructure presented a major challenge, hence the high number of individual partners. Similarly, referring Bangla-speaking participants for further mental health support was challenging as there are only a few Bangla-speaking psychologists and one psychiatrist, most based in Western Sydney. 

In building on community strengths (an assets-based community development model [40]) and existing trusted relationships to address unmet needs, the Program both leveraged and created social capital [41,42]. Julian King, a New Zealand evaluator with an interest in value for money, has introduced the term “relational efficiency” to capture the relationships, communication and trust which constitute the glue that enables programs to work efficiently, and without which resources are squandered [43]. Trust underpinned the connections between CALD community members and community partner organisations; community partners and the SESLHD Multicultural Health Team; and group participants and program providers. Trust is vital when reaching out to vulnerable groups and working with under-served communities [44]. Globally, the response to the COVID-19 pandemic highlighted the importance of community engagement and trust building to reach migrant and refugee communities [45,46,47]. In Australia, health services relied heavily upon the settlement sector and their established connections for effective community engagement [48,49]. Such collaboration and partnership should not be reserved for public health emergencies but employed, proactively, as a feature of ongoing service delivery to CALD communities. Our community partner informants also spoke about safety. When people feel safe, they can share their vulnerabilities and speak openly about their concerns in front of others without fear of judgment or rejection. A sense of safety and security facilitates learning and personal skills development [50,51].

From its inception, the CALD Mindfulness Program has benefited from strong organisational and clinical leadership. All informants spoke highly of the clinical lead who fulfilled multiple roles (program developer and coordinator, partner liaison, group facilitator, trainer and mentor), and she was consistently identified as one of main contributors to program success. At the same time relying on a “clinical champion” carries risks for sustainability. Informants emphasised the need to train more facilitators and co-facilitators, and to develop similar in-language resources for other migrant and refugee communities. This will be more challenging for new and emerging communities with limited community infrastructure and mental health expertise. Bilingual and other health professionals have an important role to play in meeting community needs [52]. High levels of somatisation, particularly in people with refugee-like backgrounds, have led to calls for training general practitioners in trauma-informed and culturally-responsive mental health care, and the use of interpreters [53].

In extending the program evaluation to encompass community partner perspectives we have learned—and relearned—much. What worked well was collaborative partnerships with organisations and individuals that were characterised by respect, trust and a shared sense of purpose. Community partners had a strong commitment to promoting community wellbeing and were keen to support the mental health initiative. Both the Multicultural Health Team and the community partners took the process of community engagement seriously, acknowledging the complexity and dynamic nature of communities and demonstrating flexibility in their response to ongoing and emerging mental health needs. Focussing on wellness and wellbeing, rather than a clinical condition or diagnosis, avoided the stigma associated with mental disorders. Adaptation to an online stress reduction program during the COVID-19 pandemic was welcomed. Positive outcomes led to stronger engagement and demand for more programs. 

Opportunities for improvement include a greater focus on equity and inclusion. It is also important that the social determinants of health, such as unemployment and secure housing, are addressed, as this ensures improvements in physical and mental health and reduction in health inequity are more likely to occur [29,54]. Refugees and asylum seekers have specific needs related to uncertainty surrounding their immigration status, as well as a lack of trust in authority figures and concerns about confidentiality [39], nevertheless the face-to-face mindfulness group delivered with the NSW Refugee Health Service was very popular. Community partner informants were critical of the geographical restrictions imposed by the current funding model. For Australians from CALD backgrounds, understanding of community is often tied to shared experiences, practices, language, beliefs and history, rather than a suburb or set of postcodes [29]. However, this approach can lead to homogenisation of what is an extremely diverse population [55]. An approach that recognises and responds to intersectionality (the range of shifting identities that people may embody and mobilise and structural power differentials) is required [55,56].

Funding from CESPHN made it possible to broaden the reach of the Program beyond the Arab speakers for whom it was originally designed to include Bangla speakers, and to expand the range of partners. Ongoing funding has enabled the development and delivery of programs in other community languages, and a growing suite of audio and video resources to support mindfulness skills development for people who are unable to attend the programs. Adequate resourcing is vital for sustainability and to ensure community partners are not overburdened. The Program, including development of in-language resources and delivery of the group programs, is both people- and time-intensive. In scaling up it will be important not to underestimate what Kavanagh and colleagues have labelled “soft infrastructure”, and which includes trust and hope, together with local knowledge and community venues that have both instrumental and symbolic value [57].

The challenges raised for policy makers are threefold. First, to reconcile the system’s reliance on geographic catchment areas for program funding and service delivery with communities’ sense of identity and social networks, and expectations that services will broadly be available to community members, particularly those living in adjacent postcodes. Second, to address the shortcomings of year-to-year, short-term funding cycles which undermine workforce stability, relational efficiency and trust in public services. Comprehensive, long-term planning based on communities’ health and social needs is required. Third, to systematically invest in soft infrastructure to support the delivery of health programs to communities experiencing disadvantage. It is common for community organisations to receive funding from a range of federal, state and local government departments for narrowly defined programs, with little room for discretionary expenditure. Through goodwill and small amounts of top-up funding, they extend their work to engage with initiatives such as the one described in this article. Their deep commitment to improving the health and wellbeing of the community they serve must be more fully recognised and remunerated.

We recognise that the CALD Mindfulness Program is only one of a growing number of psychological interventions that have been shown to improve mental health and wellbeing in migrant, refugee and asylum-seeker populations [58,59,60]. Other studies conducted among Arabic-speaking adults in Western countries support the use of culturally tailored interventions based on cognitive behaviour therapy (often including a trauma-focused component), problem solving and mind–body techniques such mindfulness, meditation and deep breathing [61,62,63,64,65,66]. Technology-based interventions are proving useful [61,62,63,64,65]. A pilot study of Mindfulness Training for Primary Care for Portuguese migrants in the United States (MTPC-Portuguese) found it to be feasible, acceptable and culturally appropriate, with statistically significant reductions in depression and anxiety symptoms [67]. 

This study has strengths and limitations. The close involvement of research team members in the group program delivery facilitated community partner recruitment and insider knowledge supported interpretation of the study findings; we consider this a strength. Purposive sampling, with participation from all but one of the selected potential informants, combined with a rigorous team-based approach to data analysis using the RADaR technique and a careful audit trail, increased the trustworthiness of the findings. Data collection was undertaken by a project officer recruited for this purpose. While being interviewed by someone familiar with the program undoubtedly assisted with establishing rapport, social desirability is an issue. Among other limitations, we asked informants to consider the Program as a whole and did not attempt to distinguish between the three components (group programs, in-language resources and workforce capacity building), nor the different group delivery formats. Most informants were members of the Arabic community, reflecting the progressive rollout (to the Arabic then the Bangla, and then to other CALD communities). Finally, all the mindfulness and stress management groups were conducted in metropolitan Sydney.

## 5. Conclusions

The CALD Mindfulness Program has considerable potential for supporting mental health and wellbeing in a multicultural nation like Australia. This community-based in-language MBI, in its various forms and formats, overcame many of the barriers to accessing timely mental health care for Arabic and Bangla communities in the CESPHN region. The initiative had a beneficial impact on group participants, program providers, partner organisations and the broader community. Community engagement utilising a strengths-based approach was central to success. Location at community venues facilitated delivery of a culturally tailored evidence-based intervention and provided a soft entry point to the health system and other services. For group mental health programs both trust and safety are necessary. Relationships must be nurtured, and projects adequately resourced so that partner organisations and program providers are not overburdened and sections of the community are not overlooked.

## Figures and Tables

**Table 1 ijerph-20-06608-t001:** Community partner informant characteristics.

ID	Partner Arrangement	Position	Informant Language	Group Language	Program Format			Informant Role		
					Face-to-Face 5 Weeks	Online 4 Weeks	Online One-Off	Support *	Co-Facilitate	Facilitate
01	Organisation	Community worker	Arabic	Arabic, Bangla	✕			✕		
02	Organisation	Community worker	Nepali	Arabic, Bangla, Nepali, English **	✕	✕	✕	✕	✕	
03	Organisation	Community worker	Arabic	Arabic	✕			✕		
04	Organisation	Community worker	Arabic	Arabic	✕			✕		
05	Individual	Community worker	Bangla	Bangla	✕	✕	✕	✕	✕	
06	Individual & Organisation	Community worker	Bangla	Bangla	✕		✕		✕	
07	Individual	Clinician	Arabic	Arabic			✕	✕		✕
08	Organisation	Community worker & Clinician	English	Arabic	✕			✕		
09	Individual	Clinician	Bangla	Bangla	✕		✕			✕
10	Organisation	Community worker	Arabic	Arabic, Bangla	✕	✕	✕	✕	✕	
11	Organisation	Community worker	Arabic	Arabic, English **	✕	✕	✕	✕	✕	
12	Organisation	Community worker	Arabic	Arabic, English **	✕		✕	✕	✕	
13	Organisation	Community worker	Bangla	Bangla	✕		✕	✕	✕	
14	Organisation	Community worker	Arabic	Arabic	✕		✕	✕	✕	
15	Organisation	Community worker	Arabic	Arabic	✕	✕	✕	✕	✕	
16	Organisation	Community worker	Arabic	Arabic	✕	✕	✕	✕	✕	

* Including promotion, recruitment, hosting face-to-face groups (organising venue, refreshments, childcare) and follow up with participants between sessions. ** One-off online session for staff.

## Data Availability

The data sets are not publicly available as they contain information that could potentially re-identify individuals, but are available from Lisa Woodland upon reasonable request and with relevant ethical approval. Program materials are available from Hend Saab.

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
