# Peer review of "“It Is Definitely a Good Program for Everyone from Every Community”: A Qualitative Study of Community Partner Perspectives on the Culturally and Linguistically Diverse (CALD) Mindfulness Program"

_ijerph, 2023, doi:10.3390/ijerph20166608_

Round 1
Reviewer 1 Report
Couple of remarks:
1. In the title of the paper is lacking information about qualitative character of the study.
2. Not enough information about CALD programme in the part "Material and method", too much in the discussion.
3. In the discussion more information about other simmilar programms needed
Best regards
Reviewer 2 Report
The project described in the article appears to be interesting. My main objection is the vague description of the mental health problems that are supposed to be alleviated/addressed among the highlighted groups. It is difficult to assess the article's argument that it has been a successful project for the involved parties without knowing anything about the problems to be solved. Are there culture-specific issues among Arabic- and Bangla-speaking individuals? Now the article seems like a biased presentation from some of the project participants who have an interest in promoting the project as successful. However, as mentioned, in order to evaluate this, the authors must provide examples of successful cases.
Reviewer 3 Report
Thank you for the opportunity to review this excellent article.
OVERALL
I believe that it is relevant to a broad, cross-field readership for the following main reasons: i. Topicality of the issue; ii. Originality with which the issue was treated; iii. Use of a "mixed-methods" methodology (quantinative and qualitative ) which, especially in papers of this type provides an important enrichment of the information derived.
However, I believe that minor observations (minor revisions), which Authors are encouraged to address, could further improve the quality of the manuscript. This relates especially to the underlying theoretical back-ground, the enrichment of which would even more prominently highlight the paper's relevance within the current medical literature.
IN DETAIL:
-Title. I encourage the Authors to find a way to highlight already in the title the focus devoted to migrants/refugees/asylum seekers. This would make the title more informative and oriented toward the content of the study.
-Keywords. I encourage the Authors to add "asilum seekers," "interpersonal relations," and "demoralization" (please also see one of the comments below). These terms, conceptually, are among the core of the paper. Moreover, they are widely used in online bibliographic searches, they can therefore help disseminate the text, make it usable by other Authors, and eventually even be cited in turn.
- Discussion. Just two observations. 1) I think it is also important to mention other types of strategies dedicated to these specific populations, including their effectiveness or otherwise. The Authors in the following paper (this is a qualitative study) will find one type of strategy, but also--in the introduction--a brief review of these strategies (rather rare on this topic). I encourage them to consult and use it: "Meaning-centered therapy in Ukraine's war refugees: An attempt to cope with the absurd?" Front Psychol. 2022 Dec 22;13:1067191. doi: 10.3389/fpsyg.2022.1067191; 2) I was deeply impressed by the participants' responses transcribed by the Authors, which amazingly largely express the sub-components of an important, intriguing, and increasingly emerging construct: namely the psychological construct of Demoralization (to be introduced in the next edition of the DSM). This construct was specifically developed in contexts of non-psychiatric patients but who had suffering related to environmental or somatic factors. Indeed, in this particular population, it is often difficult to find all the clinical criteria for making a diagnosis of depression according to DSM-5, yet psychic suffering exists. Among the forefathers in the development and application of this construct are Clarke and Kissane, whose pivotal article is: "Demoralization: its phenomenology and importance. Aust N Z J Psychiatry. 2002 Dec;36(6):733-42. doi: 10.1046/j.1440-1614.2002.01086.x." The sub-components of the definition of Demoralization are: existential despair, hopelessness, helplessness, and loss of meaning/purpose in life and dysphoria. Feelings that I think are widely experienced by these populations, as evidenced by the transcripts. And the consequences of Demoralization can be severe, including the occurrence of suicidality. That's why, from both theoretical and clinical perspectives, I think it's important for clinicians to not just consider suicidal risk only when associated with depression/anxiety, but to consider it from this more inclusive perspective. In this context, I invite the authors, in addition to the over-mentioned article of Clarke and Kissane, to also consult and use "Demoralization in suicide: A systematic review." J Psychosom Res. 2022 Jun;157:110788. doi: 10.1016/j.jpsychores.2022.110788. In it, the authors will find-especially in the introduction-a recent "mini-review" on the history of this construct. In addition, it highlights-as mentioned before-that demoralization (and not only depression in the strict sense) can result in a severe impact on mental health, up to the entire spectrum of suicidal manifestations. In the same context, interpersonal relationships play a key role in integration (in addition two sub-components of the definition of demoralization are feeling of helplessness from entourage and loss of meaning in life/purpose). For this reason, I propose this other text, "Meaning in Life and Demoralization Constructs in Light of the Interpersonal Theory of Suicide: A Trans-Theoretical Hypothesis for a Cross-Sectional Study." Psychol Res Behav Manag. 2020 Oct 29;13:855-858. doi: 10.2147/PRBM.S279829.
Minor editing (also to avoid eventual redundancies)
Reviewer 4 Report
I almost didn't review this paper because of the title and abstract, but your paper was great. I enjoyed reading your introduction, your positionality, and generally seeing strong reporting throughout your paper. It's funny that evaluations are so challenging to publish sometimes that I also try to make it less explicit in my papers - I'm assuming it was intentional on your part to not have it in the title and abstract. Anyway, please see below for my comments:
1. Title. I would strongly suggest amending your title to mention this is a qualitative study, the topic, and the study population. These may help attract readers. For instance “It is definitely a good program for everyone from every community”: A qualitative study exploring community partner perspectives on a Culturally and Linguistically Diverse mindfulness program in Sydney, Australia. I understand this is wordy, but it may help with indexing and searching once published. You could remove 1-2 elements I suggested, but my broader point of having it be more detailed stands.
2. Introduction. Change "incorporates core health promotion actions" to "considers key socioecological levels of influence described in the Ottawa..."
3. Introduction. Add a "." after the final sentence. Also, your study aim is good but you could also add what methods you use "This current study aimed to follow qualitative methods and semi-structured interviews to explore community..."
4. Methods. This is definitely a stylistic preference, but could you round up/down 40.7%, "36.8%" and 6.2% to 41%, 37% and 6%? It can make it easier for the reader to pass through the information presented.
5. Methods, 2.4.1. I'm familiar with both appreciative inquiry and socioeocological approaches, but I think some readers may not be. I think it would be nice to add a few sentences describing what they are for the uninformed reader.
6. Methods, 2.4.4. Could you move Box 1 to the Supplementary Material. Then, in the main text where you do cite "See Box 1" you could replace with "(e.g., How the program has or hasn’t met the wellbeing needs of the community? What do you think is required to sustain such a program in the community?). The full copy of the interview guide can be accessed in the Supplementary Material."
7. Methods 2.4.5. This is interesting because you cite Braun and Clarke's thematic analysis yet you choose to call it semantic analysis while also using RADaR. On top of that, you're using an appreciative inquiry and socioecological lens. Understandably, there's a lot going on here making it challenging to follow. Based on Watkins' RADaR approach, it looks like the data reduction technique also results in the generation of themes (https://journals.sagepub.com/doi/pdf/10.1177/1609406917712131). Instead of calling it semantic analysis, I would revise it to something like "We borrowed elements from thematic analysis' semantic approach to coding to support the application of the RADaR technique." Next, I would also add a few sentences on how your themes were generated using RADaR. Lastly, could you please add where you deductively derived codes from since you mention "combined inductive and deductive approaches"?
8. Results 3.2.4. I think you forgot to italicize your quotes here.
9. It's refreshing to review submissions that actually discuss researcher reflexivity. Well done.
10. Results, 3.4. The formatting at the end of this section uses a different font. Could you please fix this?
11. Great discussion section, maybe a tad bit on a longer side but then again that's just me being picky. Many of the elements resonated with me (e.g., sustainable funding, equity).
12. Strengths and limitations. Instead of "reduced the possibility of researcher bias", you could mention how a combination of this, RADaR, and audit trail strengthened the credibility or trustworthiness of your findings.
13. Strengths and limitations. I wonder if you comment on something related to the COVID-19 pandemic here since many of the programs were run during the pandemic.
14. Abstract. Circling back to your abstract, I think it may help increase readership and citations if you revised it completely. It felt really messy (please don't take this the wrong way, clearly your paper and writing is strong as expressed in the submission), with the exception of the first 2 sentences touching on the background. In terms of actionable changes, I would suggest adding "RADaR" and elements from thematic analysis were used, explicitly stating "the objective of this study was to explore community" instead of "The article reports", and listing the major themes/findings in 1 sentence.
Round 2
Reviewer 2 Report
Methodologically, you have presented a well-conducted study. However, I must unfortunately discourage publication as I do not observe sufficient changes in the content. You formulate several claims, including those related to the significance of cultural sensitivity, but fail to demonstrate how this operates in practice. What cultural differences emerged in your study that are important for someone wishing to work with mindfulness to be aware of?
